# Machine-Learning-Based Calibration of Temperature Sensors

**DOI:** 10.3390/s23177347

**Published:** 2023-08-23

**Authors:** Ce Liu, Chunyuan Zhao, Yubo Wang, Haowei Wang

**Affiliations:** 1College of Life Sciences, Fujian Agriculture and Forestry University, Fuzhou 350002, China; celiu@iue.ac.cn; 2Key Laboratory of Urban Environment and Health, Institute of Urban Environment, Chinese Academy of Sciences, Xiamen 361021, China; cyzhao@iue.ac.cn; 3School of Big Data & Software Engineering, Chongqing University, Chongqing 400044, China; wyblalala@gmail.com

**Keywords:** temperature sensor, calibration, artificial neural network (ANN), accuracy, stability, linear regression, polynomial regression, overfitting

## Abstract

Temperature sensors are widely used in industrial production and scientific research, and accurate temperature measurement is crucial for ensuring the quality and safety of production processes. To improve the accuracy and stability of temperature sensors, this paper proposed using an artificial neural network (ANN) model for calibration and explored the feasibility and effectiveness of using ANNs to calibrate temperature sensors. The experiment collected multiple sets of temperature data from standard temperature sensors in different environments and compared the calibration results of the ANN model, linear regression, and polynomial regression. The experimental results show that calibration using the ANN improved the accuracy of the temperature sensors. Compared with traditional linear regression and polynomial regression, the ANN model produced more accurate calibration. However, overfitting may occur due to a small sample size or a large amount of noise. Therefore, the key to improving calibration using the ANN model is to design reasonable training samples and adjust the model parameters. The results of this study are important for practical applications and provide reliable technical support for industrial production and scientific research.

## 1. Introduction

In recent years, China’s industrial modernization and the continuous high growth of the electronic information industry have led to a rapid rise in the sensor market. Sensors play a key role in various fields [1]. Temperature is one of the most used environmental variables and is an essential parameter in electronics, physics, chemistry, mechanics, biology, and industrial and agricultural processes. The measurement and control of temperature play a key role in ensuring product quality, improving production efficiency, saving energy, ensuring production safety, and developing the national economy. Due to the demand for temperature measuring, temperature sensors outnumber all other sensors and account for approximately 50% of all sensors.

Similarly, sensor requirements are becoming increasingly demanding, especially regarding accuracy [2]. However, temperature sensors are affected by various environmental factors and their own characteristics, such as temperature drift, hysteresis, and nonlinearity [3], causing the sensor output to deviate from the actual temperature and degrading the accuracy and stability of the sensors [4]. Inaccurate data collected with sensors [2,5] can seriously affect scientific research, industrial production, and daily life, causing problems such as the failure of scientific experiments and the production of incorrect product models. For low-cost temperature sensors, the factory accuracy is inaccurate.

Sensor accuracy depends not only on the characteristics of a sensor but also on the calibration technology [1,4,6,7]. Therefore, how to improve the measurement accuracy of temperature sensors has become a significant research topic [8]. For temperature sensors, it is essential to improve accuracy and stable measurements to ensure the accuracy of scientific research and industrial and agricultural production, as well as promote the continued development of sensors [6,9].

Various methods have recently been proposed, including linear regression, polynomial regression, and artificial neural networks (ANNs) [10,11,12,13]. Improving temperature sensor accuracy has been extensively studied [14,15]. Dias-Pereira et al. compared the data-fitting ability of traditional methods including data interpolation, least squares, polynomial fitting, and neural networks [16]. Wang et al. used the error analysis model to calibrate temperature measurements by constructing a corresponding error analysis model. Simpson et al. applied a linear correction to align the measured values with a certified and traceable reference temperature meter. Regular calibration of thermometric devices improves the accuracy of aging instruments and ensures that their sensitivity and specificity are retained in diagnosing various diseases and pathologies [17]. Wu et al. proposed a Pt100 temperature sensor calibration method based on linear regression analysis to further improve the conversion accuracy of the temperature sensor program and increase stability and reliability [18]. Miczulski et al. used Monte Carlo techniques to evaluate the uncertainty of temperature measurements and minimize the linear errors of temperature sensor processing characteristics [19].

ANN models are widely used to enhance temperature sensors’ accuracy because they capture complex nonlinearities and improve accuracy and stability [20,21,22]. The present study used an ANN model to enhance temperature sensor accuracy based on temperature data [23], and the results reveal its high accuracy of calibration. We also compared its calibration accuracy with that provided using a conventional linear regression model and a polynomial regression model. The remainder of this paper is organized as follows: Section 2 describes the materials and methods used in the experiments, Section 3 presents the experimental results and analysis, and Section 4 concludes and summarizes the advantages and limitations of these models.

## 2. Materials and Methods

### 2.1. Sensors for Calibration

Figure 1 shows the Pt100 temperature sensor that served as the calibration sensor used in this study and shows the air check bin for experimental use. Pt100 is a popular Pt thermistor temperature sensor that operates based on the temperature characteristics of Pt thermistors [24]. A Pt thermistor is a resistive element whose resistance varies with temperature as follows:(1)$$R_{t}=R_{0}(1+\alpha t),$$
where *R*_t_ is the resistance of the platinum resistor, *R*_0_ is the reference resistance, α is the temperature coefficient of the platinum thermistor, and *t* is the temperature (°C).

Since Pt thermistors have small temperature coefficients, typically approximately 0.00392 °C^−1^, an accurate temperature measurement requires a precise measurement of the resistance. A constant current source passes a constant current through the Pt100, the voltage is measured, and the resistance of the Pt100 is calculated according to Ohm’s law to obtain the temperature. Pt100s are usually available in three-wire or four-wire versions. A three-wire Pt100 has two contact resistances, which introduce errors into the circuit, whereas a four-wire Pt100 eliminates the contact resistance effect and is more accurate. In order to improve the accuracy of low-cost temperature sensors, we chose a three-wire Pt100 as the experimental sensor.

Overall, a Pt100 works based on the temperature characteristic of a Pt thermistor and determines the temperature from its resistance. Pt100s are widely used in industrial automation because of their high accuracy, good linearity, and stability.

### 2.2. Calibration Algorithms

#### 2.2.1. Linear Regression Model

A simple linear model supposes that the dependent variable Y linearly depends on the independent variable X. It is given using the mathematical formula for a line:
(2)
Y = wX + b

where Y represents the calibrated temperature value, X represents the sensor measurement temperature, and w and b are the regression coefficients.

#### 2.2.2. Polynomial Regression Model

A polynomial regression model is a machine learning algorithm for modeling nonlinear relations. It is an extension of the linear regression model, wherein the modeling of nonlinear relations is achieved by adding a high power of independent variables. In this study, a linear model was used as the polynomial regression model:(3)$$y=\beta_{0}+\beta_{1}x_{1}+\beta_{2}x_{2}+\cdots +\beta_{p}x_{p}+\varepsilon .$$

However, the independent variables *x*_1_, *x*_2_, …, and *x_p_* were no longer the original features; instead, higher powers of the independent variables were used. For example, in a quadratic polynomial regression, the independent variables *x*_1_, *x*_2_, …, and *x_p_* are one and two powers of the original features (i.e., *x*_1_, *x*_1_^2^, *x*_2_, *x*_2_^2^, …, *x*_p_, and *x*_p_^2^). In cubic polynomial regression, independent variables are 1, 2, and 3 powers of the original features, and so on. In polynomial regression, the optimal number of polynomials is determined via cross-validation. If the polynomial degree is too low, the model may not capture the nonlinear aspects; if the polynomial degree is too high, the model may over-fit the data, which reduces the possibility of generalization.

#### 2.2.3. Neural Network Model

Of the various ANNs, the Bp neural network is the most widely used and is typically used for regression and classification owing to its high degree of nonlinear fitting [25,26,27]. A Bp neural network comprises an input layer, at least one hidden layer, and an output layer. The input layer receives the input data, the hidden layer maps the input data onto a high-dimensional feature space through a series of linear transformations and activation functions, and the output layer weights and sums the output of the hidden layer to obtain the final output. During training, the network computes the output error via backpropagation and backpropagates the error layer by layer, updating the network parameters to reduce the error [28,29]. Figure 2 shows the structure of a typical Bp neural network.

As indicated, the output from neuron h in the hidden layer of the Bp neural network is
(4)$$S_{h}=f\left(\sum_{i=1}^{q}W_{ih}X_{i}-\theta_{h}\right),$$
where S_h_ is the output of the hidden layer (i.e., the neuronal output after passing through the activation function f) and refers to neuron i in the input layer, X_i_ is the input value of neuron i in the input layer, W_ih_ is the weight of neuron i in the input layer and neuron h in the hidden layer, θ_h_ is the bias of neuron h in the hidden layer, and f is the activation function for calculating the neuron output.

The input (x) is the sensor measurement temperature. After loading the data, the actual temperature and sensor-measured temperatures are stored in x and y arrays, respectively. Herein, x is an array of sensors measuring the temperature.

The output (y) is the actual temperature value after the calibration. The last layer of the model is a dense layer (Dense) with no activation function. This means that the output value is the result of the linear combination, which outputs the calibrated actual temperature value. The model takes the receiving sensor measurement temperature as an input and outputs the corresponding calibrated actual temperature value.

The output Y_j_ can be expressed as
(5)$$Y_{j}=f\left(\sum_{h=1}^{d}W_{hj}S_{h}-\theta_{j}\right),$$
where d is the number of neurons in the hidden layer, W_hj_ is the weight of neuron j in the output layer and neuron h in the hidden layer, Sh is the output of neuron h in the hidden layer, and θ_j_ is the bias of neuron j in the output layer.

The output error E can be obtained using
(6)$$E=\frac{1}{2}\sum_{j=1}^{m}{\left(t_{j}-y_{j}\right)}^{2},$$
where m is the number of neurons in the output layer, t_j_ is the target output of the sample, and y_j_ is the real output of the neural network.

A three-layer backpropagation ANN was used to calibrate the temperature sensor and a hidden layer containing 16 neurons, and, to minimize the loss function, the ReLU activation function was used to update the weight and bias of the model in each training cycle (epoch) according to the gradient of the loss function.

### 2.3. Data Source

#### Data Collection

The temperature data were collected from −50 to 140 °C. A total of three sensor data were recorded at intervals of 0.2 °C. As the temperature rose, the Pt100 temperature sensor changed its resistance. The resistance rose as the temperature rose, which explains the positive resistance coefficient. The error is calculated as follows:


E = E_Measure_ − E_actual_
(7)


## 3. Results

The linear regression model w was 0.9887 and b was −0.577. The fitting formula was
(8)Y=0.9887×X−0.577

The polynomial regression model coefficient was 0.9903 and −0.00001807, the constant term was −0.5636, and the fitting formula was
(9)$$Y=0.9903\times X^{2}-0.00001807\times X-0.5636$$

To determine the temperature error of the sensors, they measured across a temperature range of −40 to 140 °C. Figure 3 shows the temperature error of the sensors. The 3σ spread across the range of −40 to 140 °C was 0.61 °C.

Figure 3 shows a large error between the actual and measured temperatures of the three sensors when they were not calibrated. The maximum error was ±2.74 °C. The actual temperature in the figure is the temperature displayed by the experimental gas box set with the control software, and the measured temperature is the displayed value measured using the temperature sensor.

### 3.1. Calibration Based on Machine Learning Algorithms

The temperatures measured with Sensors A, B, and C were calibrated using the linear regression model, the polynomial regression model, and the neural network model, respectively. Herein, the differences between the temperatures measured with the low-cost sensors and the actual temperature were regarded as the original error of the algorithm. As indicated, calibrations using all three methods increased the accuracy and reduced the error. The neural network model produced the most accurate calibration of the three models, where Figure 4, Figure 5, Figure 6, Figure 7, Figure 8 and Figure 9 demonstrate the suitability of the proposed machine-learning-based method for temperature sensors.

To enhance the accuracy of the sensors, we used the ANN algorithm for sensor calibration, which yielded the error results plotted in Figure 4.

#### 3.1.1. Calibration Using ANN Regression

As shown in Figure 4, the maximum error decreased from 0.61 °C (3σ) to 0.167 °C (3σ), with a maximum error of less than 0.05 °C within the commonly used temperature range of 25 to 75 °C, thereby validating the effectiveness of the ANN model. To further evaluate the performance of the ANN model, two additional models were implemented in this research: linear and polynomial models. All models were trained and tested in the same environment. The results are as follows.

As shown in Figure 5, after using the ANN model, the error was obvious at approximately −50 °C, and it was relatively flat between −25 and 125 °C.

#### 3.1.2. Calibration Using Linear Regression

As shown in Figure 6 and Figure 7, from 0 to 100 °C, the effect of the model error reduction was satisfactory, and from −50 to 0 °C and 100 to 150 °C, the effect on the error was insufficient.

#### 3.1.3. Calibration Using Polynomial Regression Model

As shown in Figure 8 and Figure 9, from 25 to 75 °C, the effect of the model error reduction was satisfactory, and from −50 to −30 °C and 75 to 100 °C, the effect on the error was insufficient.

Table 1 provides a comparison of the performance of different models. It is worth noting in Table 1 that all three methods have improved, with the best performing being the ANN model, which obtained smaller error uncertainty, mean, and standard deviation.

### 3.2. Performance Evaluation

The calibration methods were evaluated based on the mean average error (MAE), mean square error (MSE), and the correlation coefficient squared (R^2^) between the temperatures measured using a calibrated Pt100 sensor and the actual temperature.
(10)$$MAE=\frac{1}{n}\sum_{i=1}^{n}|y_{actual,i}-y_{predicted,i}|$$
where n is the number of samples, y actual,i is the actual predicted value y of the ith sample, and i is the predicted value of the ith sample.
(11)$$MSE=\frac{1}{n}\sum_{i=1}^{n}{\left(y_{actual,i}-y_{predicted,i}\right)}^{2}$$
(12)$$R^{2}=1-\frac{\sum_{i=1}^{n}{\;{(y}_{actual,i}-y_{predicted,i})}^{2}}{\sum_{i=1}^{n}{\;(y_{actual,i}-\hat{y_{actual,i}})}^{2}}$$
where y_actual,i_ is the mean of the actual values.

#### Performance Index Results

To illustrate the effectiveness of the model method for improving the accuracy of temperature sensors, we evaluate the index performance of the proposed method.

The Table 2 provides the performance comparison of the different models. It can be seen from the table that the three models were effective in improving the accuracy of the sensors. The MSE, MAE, and R^2^ of the ANN model were more accurate and the improvement in the accuracy of the temperature sensors was even greater. Therefore, it is effective to consider using the ANN model to improve the accuracy of temperature sensors. Among the three methods, the ANN model performed the best.

## 4. Discussion

The widespread use of temperature sensors has led to inexpensive sensors that are less accurate than one would expect, so improving the accuracy of temperature sensors was the main goal of this paper. The efficacy of calibration methods was evaluated by calibrating temperature sensors from −50 to 150 °C. This study used linear regression, polynomial regression, and a neural network model to calibrate the sensors.

The advantages of calibrating temperature sensors using linear regression are that these models are simple to understand, rapid to calculate, and suitable for linearly related data. However, linear regression is only applicable to linear data and may be susceptible to over-fitting. The calibrations of sensors A–C using linear regression produced the slopes w = 0.9951, 0.9903, and 0.9887, respectively, and the y-intercepts were −1.081, −0.8312, and −0.5769, respectively. MSE, MAE, and R^2^ were used to quantify the quality of calibration. The MAEs of sensors A–C after calibration using linear regression were 0.134, 0.2009, and 0.218, respectively; the MSEs of sensors A–C after calibration using the linear regression model were 0.03476, 0.07067, and 0.0713, respectively; and the R^2^ of sensors A–C after calibration using linear regression were 0.999 977 2, 0.997 66, and 0.999 974 9, respectively. The results demonstrate that calibration using linear regression significantly improved the sensors’ performance compared with that before calibration. Additionally, the MSE, MAE, and R^2^ of the training and testing sets revealed no over-fitting.

The advantage of calibrating temperature sensors using polynomial regression is that more complex data can be fit, and the complexity of the model can be controlled by choosing polynomials of differing degrees. However, polynomial regression may be susceptible to over-fitting. Likewise, MSE, MAE, and R^2^ were used to quantify the measurement accuracy. The MAEs of sensors A–C after calibration using polynomial regression were 0.128, 0.166, and 0.207, respectively; the MSEs of sensors A–C after calibration using polynomial regression were 0.023, 0.0438, and 0.0589, respectively; and the R^2^ of sensors A–C after calibration using polynomial regression were 0.9999787, 0.9999839, and 0.9999789, respectively. The results demonstrate that calibration using polynomial regression significantly improved the sensor accuracy compared with that before calibration. Additionally, the MSE, MAE, and R^2^ of the training and testing sets revealed no over-fitting.

Calibrating temperature sensors using the neural network model has the advantage that more complex nonlinear data can be fit, and the model performance can be improved by adjusting the network structure and hyperparameters. However, the neural network model requires more data and computational resources and more experience in selecting network structures and parameters. The MSE, MAE, and R^2^ were used to quantify the quality of calibration. The MAEs of sensors A–C after calibration using the neural network model were 0.0992, 0.1078, and 0.0916, respectively; the MSEs of sensors A–C after calibration using the neural network model were 0.0196, 0.0333, and 0.0354, respectively; and the R^2^ of sensors A–C after calibration using the neural network model were 0.9999873, 0.9999886, and 0.9999874, respectively. The results demonstrate that calibration using the neural network model significantly increased the sensor accuracy compared with that before calibration. Additionally, the MSE, MAE, and R^2^ of the training and testing sets revealed no over-fitting.

First, according to all the performance metrics, the neural network model performed the best, and linear regression performed the worst. Moreover, linear regression is applicable to linear data, polynomial regression is applicable to complex data, and the neural network model is applicable to nonlinear data. Thus, a model’s performance is related to its complexity and data quality. The neural network model and polynomial regression are both more complex than linear regression and can better accommodate the nonlinear characteristics in the data. Nevertheless, these complex models can be overfitted if the data are of poor quality or noisy. Additionally, neural network models require longer training times, so time costs need to be considered in practical applications.

Like all studies, this study has its limitations. First, only a single feature was used for training and testing, so these findings may not be applicable to calibrating temperature sensors with other features. In addition, only three models were considered, whereas other more complex or simpler models can be used for sensor calibration. In future research, more models and datasets should be explored to further optimize the calibration of temperature sensors.

The results show that the neural network model is the optimal method for calibrating temperature sensors. However, time costs and data quality must be considered. This study provides valuable insights for future research into optimizing temperature sensors.

## 5. Conclusions

Conventional calibration methods for temperature sensors, including linear regression and polynomial regression, have disadvantages, such as low accuracy. This study proposed a calibration method based on the neural network model. To evaluate the performance of the neural network model, three sensors were used to collect 829, 891, and 924 temperature measurements between −50 and 150 °C. Herein, all the conditions were unified, and the temperature was set in the air check bin to ensure that any error originated from the sensors themselves. The three sensors were then calibrated, and the accuracy was quantified using the MAE. After calibration, the sensor accuracy improved significantly. The three models all improved the accuracy of measurement, but the neural network model produced the most accurate sensor measurements. The results show that different sensors can be calibrated using this method, and the differences in sensors reside in different features, which can be accommodated by training the neural network model with adjusted weight and bias to improve the measurement accuracy.

This study thus calibrated temperature sensors using linear regression, polynomial regression, and a neural network model and compared the accuracy of the resulting measurements. The following conclusions can be drawn.

The measurement error of the temperature sensor was relatively stable at approximately 1 °C from 0 to 100 °C and increased to approximately ±2 °C for the ranges −50, −40 °C, and 120–150 °C. The maximum differences in the measurements of the three sensors before and after calibration were 2.74 and 1.71 °C, respectively.

The neural network model provided an optimal calibration of the temperature sensors used in this study. Specifically, calibration using the neural network model produced a low MAE and MSE and a high decision factor R^2^, suggesting that the neural network model calibrates sensor readings more accurately and provides a better fit between the actual temperature and the measured temperature.

Linear regression can also be used to calibrate temperature sensors. Calibration using linear regression produced a relatively high R^2^, but its MAE and MSE were slightly inferior to those of polynomial regression, suggesting that the linear regression model better explains the linear relation between temperature and sensor readings. However, for more complex nonlinear relations, the effect may be reduced. Additionally, polynomial regression was used to calibrate the temperature sensors, and the experimental results show that polynomial regression produced more accurate temperature sensors. However, based on the MAE and MSE, calibration using polynomial regression produces slightly less accurate measurements than calibrating using the neural network model, indicating that although polynomial regression better explains the nonlinear relation between temperature and sensor readings, it may be slightly less interpretable than the neural network model. Calibration using the proposed neural network model works better than using the other methods, with the calibration accuracy for the three sensors being 92.24%, 91.8%, and 91.9%, respectively, which is better than those of polynomial regression (90.06%, 87.3%, and 81.8%) and linear regression (89.57%, 84.6%, and 80.8%).

In summary, the neural network model is ideal for calibrating temperature sensors. When more complex nonlinear relations need to be explained, try using polynomial regression. Linear regression is suitable for simple linear relations.

## Figures and Tables

**Figure 1 sensors-23-07347-f001:**
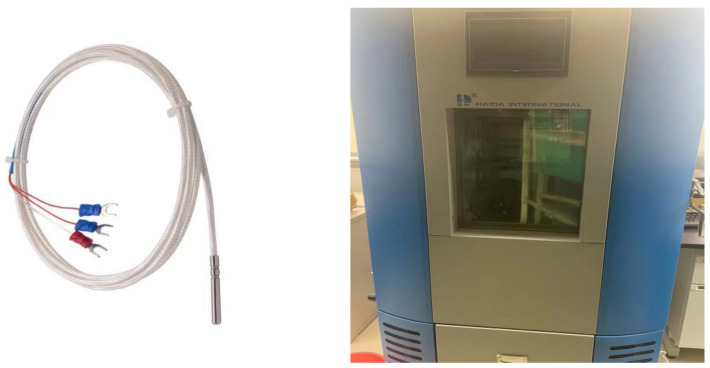
Pt100 temperature sensor and air check bin used in this study.

**Figure 2 sensors-23-07347-f002:**
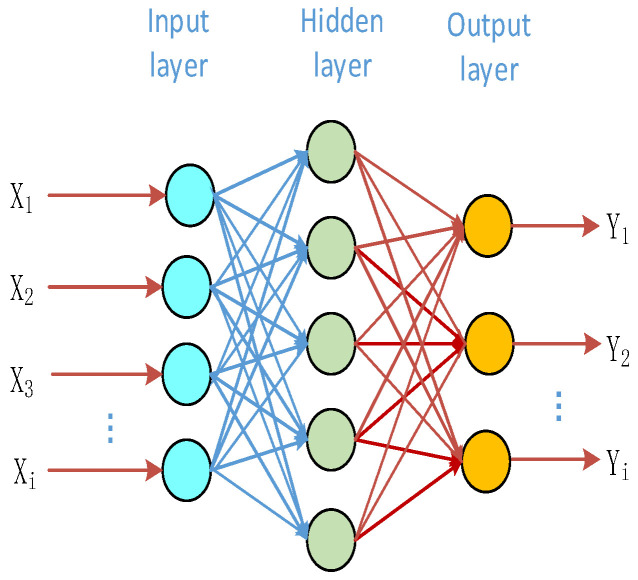
Structure of a typical neural network model.

**Figure 3 sensors-23-07347-f003:**
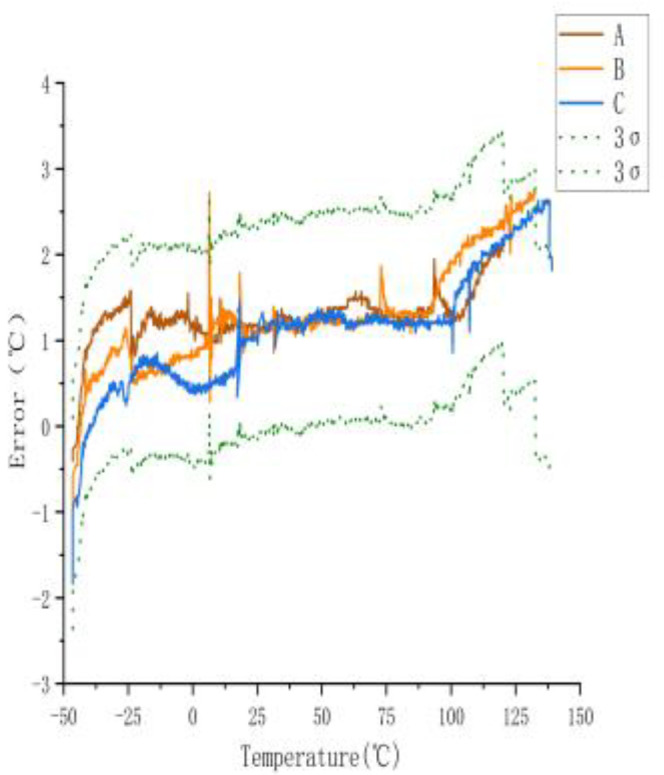
Measurement error of uncalibrated sensor.

**Figure 4 sensors-23-07347-f004:**
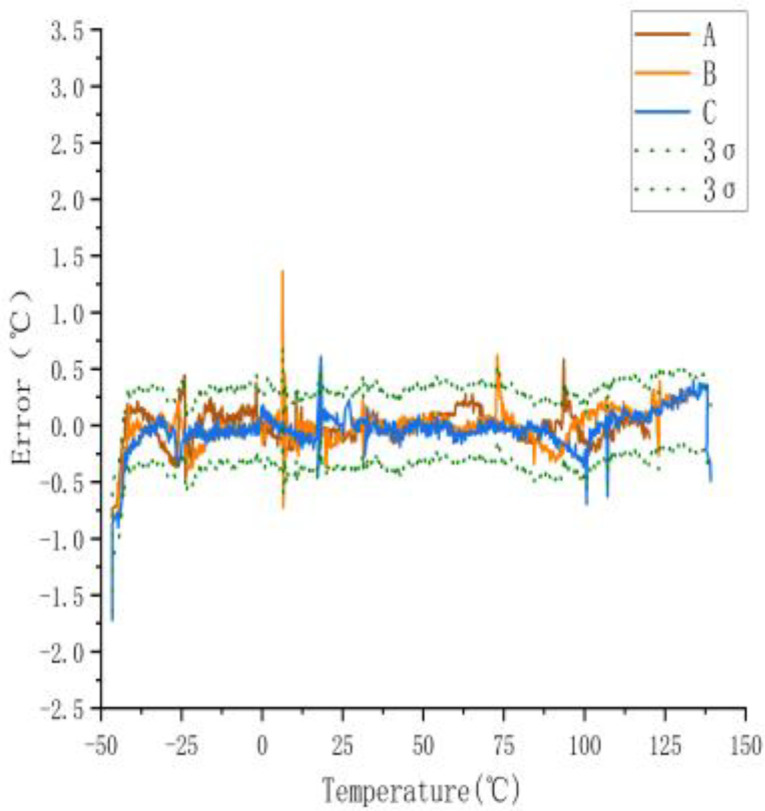
Errors after using ANN model.

**Figure 5 sensors-23-07347-f005:**
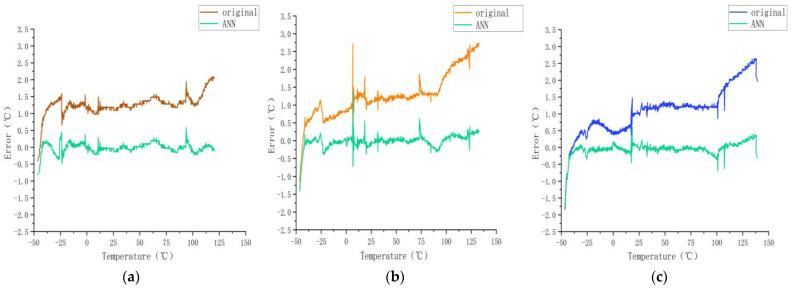
Calibration results for sensors (**a**) A, (**b**) B, and (**c**) C obtained using the neural network model.

**Figure 6 sensors-23-07347-f006:**
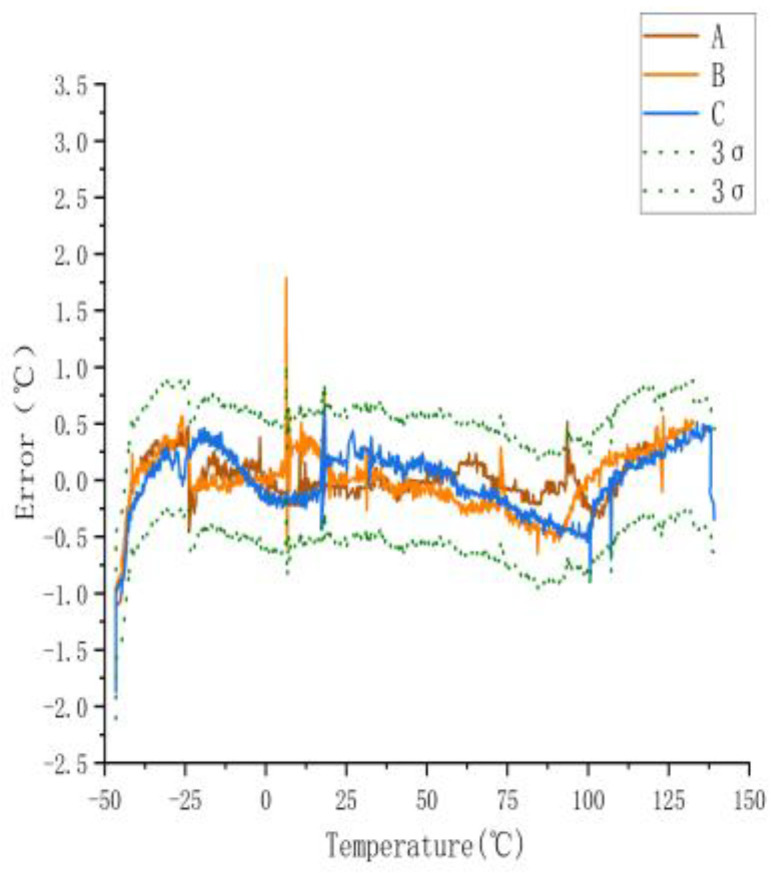
Errors after using linear regression model.

**Figure 7 sensors-23-07347-f007:**
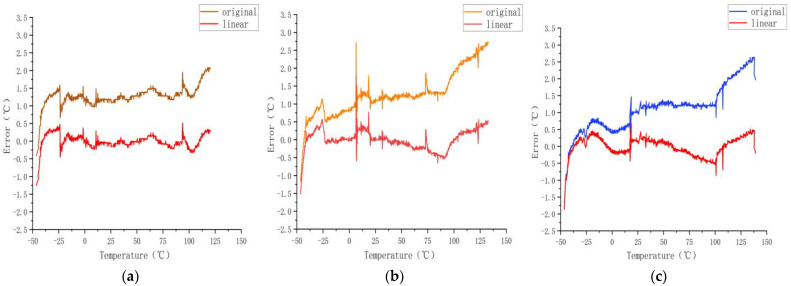
Calibration results for sensors (**a**) A, (**b**) B, and (**c**) C obtained using the linear regression model.

**Figure 8 sensors-23-07347-f008:**
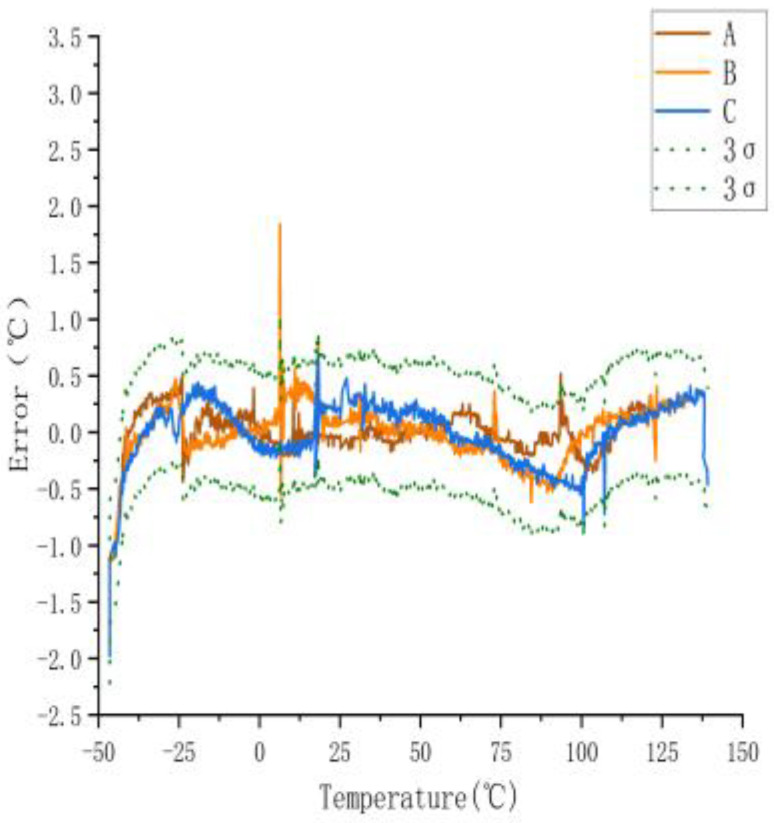
Calibration results for sensors obtained using the polynomial regression model.

**Figure 9 sensors-23-07347-f009:**
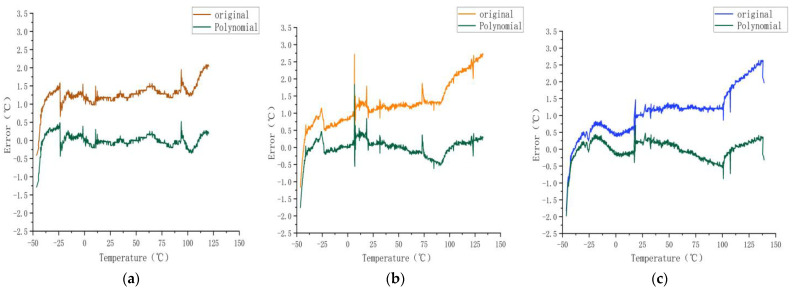
Calibration results for sensors (**a**) A, (**b**) B, and (**c**) C obtained using the polynomial regression model.

**Table 1 sensors-23-07347-t001:** Comparison between the temperature sensor performances.

Sensor	Method	Inaccuracy	Mean Value	Standard Deviation
A	Original	±0.529	1.28	0.27
	ANN	**±0.267**	**−0.0104**	**0.137**
	Linear	±0.365	−0.0007	0.186
	Polynomial	±0.364	0.0034	0.1856
B	Original	±1.16	1.289	0.59
	ANN	**±0.322**	**−0.0089**	**0.165**
	Linear	±0.521	0.0072	0.266
	Polynomial	±0.481	0.0007	0.246
C	Original	±1.3	1.124	0.66
	ANN	**±0.317**	−0.0317	**0.162**
	Linear	±0.523	0.0035	0.267
	Polynomial	±0.514	0.001	0.262

Bold mean the most effective numbers.

**Table 2 sensors-23-07347-t002:** Evaluation of each method index of the temperature sensors.

Sensor	Method	MSE	MAE	R^2^
A	Original	1.731	1.284	0.99898
	ANN	**0.0196**	**0.0992**	**0.9999873**
	Linear	0.03476	0.134	0.9999772
	Polynomial	0.026	0.128	0.9999787
B	Original	2.0506	1.314	0.999323
	ANN	**0.0333**	**0.107766**	**0.9999886**
	Linear	0.070668	0.2009	0.9999766
	Polynomial	0.0438	0.166	0.9999839
C	Original	1.686	1.136	0.99396
	ANN	**0.0354**	**0.0916**	**0.9999874**
	Linear	0.0713	0.218	0.9999749
	Polynomial	0.0589	0.207	0.9999789

Bold mean the most effective numbers.

## Data Availability

Not applicable.

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
