# Peer review of "Machine-Learning-Based Calibration of Temperature Sensors"

_sensors, 2023, doi:10.3390/s23177347_

Round 1
Reviewer 1 Report
The manuscript entitled "Machine-Learning-Based Calibration of Temperature Sensors" proposes an approach to enhance temperature calibration accuracy through the utilization of an Artificial Neural Network (ANN) model. Despite its promising premise, the current form of the manuscript contains significant flaws that raise concerns about its suitability for publication. Allow me to elaborate on the specific points of concern:
-
Throughout the manuscript, the authors discuss the concept of calibration but curiously omit any mention of traceability to primary or secondary standards. Calibration, by its nature, requires the establishment of traceability to recognized standards, such as those provided by esteemed institutions like the National Metrological Institute. The absence of such traceability calls into question the robustness and reliability of the calibration process employed.
-
An essential aspect of any reliable measurement, particularly in calibration, is the calculation and reporting of uncertainty values. The manuscript lacks this critical component, which is pivotal for establishing the credibility and accuracy of the calibration methodology. Without a thorough assessment of uncertainty, the confidence in the calibration results becomes compromised.
-
The statistical analysis presented in the manuscript is limited to basic methods, which may be inadequate to support the authors' claim of improving calibration capability. A more comprehensive and rigorous statistical analysis is necessary to bolster their assertions effectively and to provide a solid foundation for their proposed approach.
In light of these concerns, I strongly recommend that the authors reconsider their focus by removing the calibration claim and instead concentrate on enhancing the temperature measurement capability. Additionally, I advise them to delve into the details of primary and secondary standards and familiarize themselves with the traceability pyramid to improve the clarity, rigor, and overall scientific merit of their work. Addressing these issues will undoubtedly strengthen the manuscript and make it more suitable for publication consideration.
The English language is fine; however, authors should proofread the manuscript multiple times to ensure it is error-free before submitting the revised version.
Author Response
请参阅附件

Reviewer 2 Report
The article “Machine-Learning-Based Calibration of Temperature Sensors” is based on the characterization of artificial neural network (ANN) model for calibration and explores the feasibility and effectiveness 12 of using ANNs to calibrate temperature sensors. This work needs extensive changes and not much novel, it can be considered for potential publications after the following major changes:
1. References do not follow the same style. Some references, all names are capitalized like Ref [1, 5, 19, 10].
2. Sometimes after a full stop, two spaces are observed, like ref 2, while in some cases, there is no spacing, like ref 3 in the introduction section.
3. Figures 1 and 2 can be combined.
4. Equations should be mentioned before or after in theory.
5. In Eq (1) both alpha are different; recheck the subscripts too.
6. Authors mention 4 contact PT100 is better than 3 contacts, but in Figure 1 they presented 3 contacts.
7. After Eq (2), there is Yi which is not used before or after what is the purpose of writing it.
8. Figures are quite blurred and should be of good quality graphs should use symbols with lines so that during print, researchers can differentiate, as printed papers are usually not in colored form.
9. Most results look superficial and may need more explaination, as sections 3.1.2 and 3.1.3 have graphs but no theory to support them.
10. Figures 8-16 can be combined as the same x-axis and parameters. Only the y-axis is changing.
11. Highlight the novelty, as this work has already been explored a lot.
Reviewer 3 Report
This study compared the error correction algorithms of Pt100 temperature sensors. The manuscript needs further refinement.
1. What is the reason for the Pt sensor error due to the unclear introduction of the problem to be studied in the manuscript?
2. What method was used to obtain actual temperature data? What do the actual temperature and measured temperature in Figure 4 refer to respectively? What is the calculation formula for error?
3. What is the difference between the linear regression model and the polynomial regression model in this case? What are the specific values of coefficients in each model? The calculated fitting formula needs to be supplemented.
4. What are the inputs and outputs of the ANN model?
5. The original data curve in Figure 7 is inconsistent with the original data curve in Figure 5 and Figure 6. Is the data reliable?
6. It is recommended to use a table instead of a bar chart from Figure 8 to Figure 16 to compare the values of the MAE, MSE, and R2 parameters
Author Response
请参阅附件

Round 2
Reviewer 1 Report
The authors have significantly improved the manuscript, addressing previous concerns. The revised version is now acceptable for acceptance in its current form.
Reviewer 2 Report
Can be accepted. But not much novel.
Reviewer 3 Report
The manuscript has been revised carefully and can be accepted by the journal.